# Large oscillatory thermal hall effect in kagome metals

Dechen Zhang [1], Kuan-Wen Chen [1], Guoxin Zheng [1], Fanghang Yu[2], Mengzhu Shi[2], Yuan Zhu [1], Aaron Chan[1], Kaila Jenkins [1], Jianjun Ying [2], Ziji Xiang [1,2], Xianhui Chen [2] & Lu Li [1] ✉

The thermal Hall effect recently provided intriguing probes to the ground state of exotic quantum matters. These observations of transverse thermal Hall signals lead to the debate on the fermionic versus bosonic origins of these phenomena. The recent report of quantum oscillations (QOs) in Kitaev spin liquid points to a possible resolution. The Landau level quantization would most likely capture only the fermionic thermal transport effect. However, the QOs in the thermal Hall effect are generally hard to detect. In this work, we report the observation of a large oscillatory thermal Hall effect of correlated Kagome metals. We detect a 180-degree phase change of the oscillation and demonstrate the phase flip as an essential feature for QOs in the thermal transport properties. More importantly, the QOs in the thermal Hall channel are more profound than those in the electrical Hall channel, which strongly violates the Wiedemann−Franz (WF) law for QOs. This result presents the oscillatory thermal Hall effect as a powerful probe to the correlated quantum materials.

The thermal Hall effect has been a powerful technique to probe the exotic nature of correlated quantum matter[1–16]. The thermal Hall effect is the thermal analog of the electrical Hall effect, which detects the transverse temperature gradient in the presence of longitudinal heat current and perpendicular magnetic field. The thermal Hall effect measured the Bogoliubov quasiparticles directly in the high-temperature superconducting cuprates[2–5] and Fe-based superconductors[6]. It measured the magnon excitations in the quantum magnets[7–10] and resolved the Majorana quantizations of the edge state in the Kitaev spin liquid candidate $\alpha$-RuCl$_3$[11,12]. On the other hand, recent progress points to the phonon origin of the thermal Hall effect in some correlated materials. For example, the Mott insulator state of cuprates is reported to establish a giant thermal Hall effect[13,14]. The latest report reveals the large phonon thermal Hall effect in quantum paraelectric material SrTiO$_3$[15].

The thermal Hall effect is much more universal than the electrical Hall effect. It is the consequence of the chirality of the carriers, whether they are fermions or bosons. The observation of these unconventional thermal Hall effects has provided many fresh insights to the exotic nature of correlated quantum matter. However, the debate has always been whether the origins of these reported effects are due to fermions or bosons[16–20]. Thus, a question needs to be answered on how to confirm whether a thermal Hall signal is a fermionic or bosonic response. For example, the thermal Hall signal in $\alpha$-RuCl$_3$ is debated to come from either phonons, magnons, or the proposed edge state fermions[1,10–12,21]. One remarkable progress is the observation of QOs in the thermal conductivity of $\alpha$-RuCl$_3$[1], which reveals the fermionic nature of the spin liquid phase because QOs are the result of Landau Level quantization for Fermi surfaces. This observation suggests a crucial potential path to separate the fermionic and bosonic thermal Hall effects by resolving the QOs. The temperature variance of the oscillatory component will further answer if the original theory based on Fermi Liquid theory for QOs in magnetization and electrical resistivity can describe the pattern of QOs in the thermal transport[1].

However, even the thermal Hall effects are most often negligibly small in real materials. The observation of QOs in the magnetothermal effect is only limited to the two elemental metals, aluminum and

¹Department of Physics, University of Michigan, Ann Arbor, MI, USA. ²CAS Key Laboratory of Strongly-coupled Quantum Matter Physics, Department of Physics, University of Science and Technology of China, Hefei, Anhui, China. ✉e-mail: luli@umich.edu

zinc[22–25]. For correlated metals, the report of transverse magnetothermal QOs is entirely missing. In this report, we chose the recently discovered Kagome metal $CsV_3Sb_5$ as a platform[26,27], in which the unusual electronic structure has already led to the demonstration of large anomalous Hall, Nernst, and thermal Hall effects[28–31], the electronic nematicity[32–35], and the pairing-density-wave phenomena in the superconducting and possible pseudogap state[36,37]. We present the first observation of the QOs in the thermal Hall effect in quantum-correlated materials. In $CsV_3Sb_5$, we were able to determine the Wiedemann–Franz (WF) ratio between the thermal Hall and electrical Hall QOs. At the ground state, the ability of charged quasiparticles to transport heat and charge is governed by the universal WF law. Violations of the WF law are typically an indication of unconventional quasiparticle dynamics, such as inelastic scattering, semimetal physics, or new phases of matter[38–44]. Indeed in $CsV_3Sb_5$, we found the low-temperature oscillation amplitude of thermal hall conductivity is enhanced by a factor of 2.5 compared with that in electrical Hall conductivity multiplied by the Sommerfeld value $L_0$ and the absolute temperature $T$, which cannot be explained by the conventional WF ratio. This strong violation of the oscillation WF law challenges the fundamental concept of Landau quasiparticles and is suggestive of an exotic correlated quantum phase.

## Results

### The longitudinal thermal conductivity and transverse thermal Hall signals measured in $CsV_3Sb_5$

The $CsV_3Sb_5$ single crystal was synthesized via a self-flux growth method similar to the previous reports[26]. The measurement was performed in the Oxford Triton200-10 Cryofree Dilution Refrigerator. The

longitudinal and transverse thermal conductivity were measured using a one-heater-three-thermometers technique (Fig. 1a). The longitudinal and transverse temperature differences $\Delta T_x$ and $\Delta T_y$ were read by Lakeshore Cryotronics RX102A thermometers. The $H$-field was varied continuously and slowly during the measurement at a stable temperature. The results were also checked to be consistent with the stepped-field method to avoid the eddy current or other heating effects. The field dependence of the measured signal was plotted as thermal and thermal Hall resistivities to prevent the error in doing the matrix inversion. (The details of the measurement are in "Methods").

As shown in Fig. 1a, $\Delta T_x$ and $\Delta T_y$ were measured at fixed $T$ with $H\|z$ and $-\nabla T\|x$. Then, the thermal resistivity matrix $\lambda_{ij}$ was obtained using the recorded temperature gradient $-\nabla_x T$ and $-\nabla_y T$ (see "Methods" for experimental setup). Figure 1b displays the longitudinal thermal resistivity $\lambda_{xx}$ as a function of the $H$-field. Around the zero field, $\lambda_{xx}(H)$ shows a sudden enhancement from the superconducting state as the sample temperature is below $T_c$ (~2.5 K). Above ~4 T, clear oscillations emerge in $\lambda_{xx}(H)$. The $T$ dependence of the thermal conductivity also reveals an interesting feature. In a metal, the total thermal conductivity $\kappa_{xx}$ is the sum of the phononic ($\kappa_{ph}$) and electronic ($\kappa_e$) contributions. Due to different scattering mechanisms of electron and phonon, $\kappa_{xx}$ is dominated by the electron at low $T$. The Wiedemann–Franz (WF) law states that in metals the WF ratio $L = \frac{\kappa_{xx}}{\sigma_{xx}T}$ is nearly the Lorenz number $L_0 \left( 2.44 \times 10^{-8} V^2 K^{-2} \right)$, where $\sigma_{xx}$ is the electrical conductivity. The electronic contribution to the thermal conductivity in $CsV_3Sb_5$ was calculated using $L_0\sigma_{xx}T$, where $\sigma_{xx}$ is measured by using the same contacts on the sample connected to the current leads (see Supplementary Fig. 2). The value of $L_0\sigma_{xx}T$ is then compared with the total thermal conductivity. Even down to lowest $T$, the total thermal

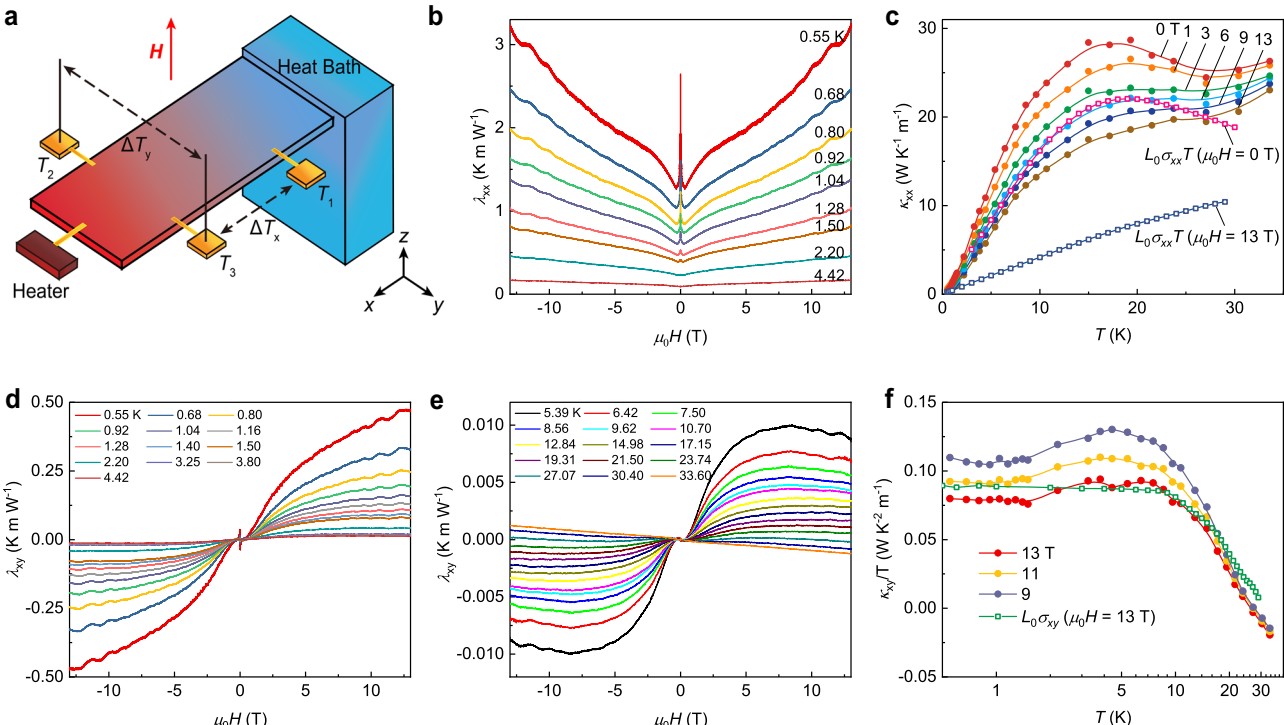

**Fig. 1 | The magnetothermal and thermal Hall signals measured in $CsV_3Sb_5$.**
**a** Sketch of the thermal Hall experiment setup. Three thermometers are connected to the sample through gold wires to measure the longitudinal and transverse temperature differences $\Delta T_x$ and $\Delta T_y$. **b** The magnetic field dependence and QOs of longitudinal thermal resistivity $\lambda_{xx}(H)$. The labeled $T$ is the average temperature on the sample. The data were recorded continuously with the $H$-field swept with sufficiently slow speed. **c** The longitudinal thermal conductivity $\kappa_{xx}$ as a function of measured $T$ in different magnetic fields $H$. The pink and blue curves with open squares are the electronic thermal conductivity at 0 T and 13 T calculated according

to the WF law, respectively. **d** The thermal Hall resistivity $\lambda_{xy}(H)$ displays large quantum oscillations at selected $T$ from 0.55 K to 4.42 K. **e** $\lambda_{xy}(H)$ above 5.39 K. In the low field region below ~30 K, the S-shaped anomalous thermal Hall signal shows up in $\lambda_{xy}(H)$. Above ~30 K, $\lambda_{xy}(H)$ shows ordinary $H$-linear behavior. **f** The temperature dependence of the thermal Hall conductivity $\kappa_{xy}/T$ at selected $H$. The green curve with the open circle is the electronic thermal Hall conductivity at 13 T multiplied by the Lorenz number $L_0 \left( 2.44 \times 10^{-8} V^2 K^{-2} \right)$. At base temperature, the employed Hall WF ratio $L = \frac{\kappa_{xy}}{\sigma_{xy}T}$ is ~ $2.2 \times 10^{-8} V^2 K^{-2}$.

conductivity is found to be greater than $L_0 \sigma_{xx} T$ (Fig. 1c). The large deviation indicates a significant contribution of phonon thermal conductivity before entering the superconducting state.

Next, we turn to examine the thermal Hall effect in CsV$_3$Sb$_5$. As shown in Fig. 1e, above 30 K, $\lambda_{xy}$ shows the $H$-linear conventional-metal pattern, indicating the carrier density is dominated by one band. Below 30 K, the thermal Hall resistivity $\lambda_{xy}$ becomes strongly nonlinear in $H$, accompanied by a sign change. The same temperature range also coincides with the onset temperature of electronic nematicity[35], and the nonlinearity in Hall and Nernst signals[28,29,31], which was attributed to the enhancement of hole mobility at low $T$[37,45]. In the low $H$ region, $\lambda_{xy}$ also shows a significant anomalous thermal Hall effect, which suggests the possible time-reversal-symmetry-breaking nature of charge order[46]. The large anomalous signal decreases as temperature increases but stays robustly at elevated temperatures up to 30 K (see Supplementary Fig. 8).

We note that the observed non-oscillating background of $\lambda_{xy}$ is still consistent with the WF law at very low $T$ (Fig. 1f). Since the transverse channel is usually free of phonon contribution, the WF ratio $L$ is directly related to the electrical and thermal relaxation time. In CsV$_3$Sb$_5$, we extract and compare the non-oscillating part $\bar{\kappa}_{xy}(T)/T$ with that of $L_0 \bar{\sigma}_{xy}$, where $\bar{\sigma}_{xy}$ is the non-oscillating background of electrical Hall conductivity. As $T$ drops, the WF ratio $L$ increases and recovers to $L_0$. Below ~15 K, the value of $\bar{\kappa}_{xy}(T)/T$ becomes quite close to $L_0 \bar{\sigma}_{xy}$. Below 10 K, both $\bar{\kappa}_{xy}(T)/T$ and $L_0 \bar{\sigma}_{xy}$ become nearly a constant. With further drop in $T$, $\bar{\kappa}_{xy}(T)/T$ slightly decreases. The WF law of the transverse non-oscillating background is further supported by the data of another CsV$_3$Sb$_5$ sample, as shown in Supplementary Fig. 9.

The large QOs also appear in the thermal Hall resistivity $\lambda_{xy}$ above ~4 T (Fig. 1d, e). Differing from the $H$-symmetric oscillation in $\lambda_{xx}$, the $H$-antisymmetric oscillation pattern in $\lambda_{xy}$ is intrinsic and does not come from the longitudinal pick-up (see Supplementary Fig. 4). By performing the Fast Fourier transformation (FFT) (Fig. 2a, b), the oscillations contain multiple orbits whose frequencies are consistent with those in the Shubnikov-de Haas (SdH) oscillation[47] and the de Haas–Van Alphen (dHvA) oscillation[48]. Among all these four compositions, the contribution from the $\delta$ orbit (frequency $F$ ~87 T) dominates when $H$ is greater than ~9 T.

### The temperature dependence of the amplitude of the oscillatory magnetothermal effect

We further investigate the amplitude of the oscillating components of the thermal resistivity $\tilde{\lambda}_{xx}$ and the thermal Hall resistivity $\tilde{\lambda}_{xy}$. The first question is what the temperature dependence of the magnetothermal oscillation amplitudes should be. According to the prediction from the Fermi liquid theory, the expected temperature dependence of the oscillation amplitude of the magnetothermal effect is different from the well-established Lifshitz–Kosevich $T$-dependence for magnetization and resistivity. Instead,

$$R_T(T) = \frac{\frac{2\pi^2 k_B T}{\hbar \omega_C}}{\sinh\left(\frac{2\pi^2 k_B T}{\hbar \omega_C}\right)} \quad (1)$$

is replaced by $R_T''(T)$, its second derivative with respect to $T$. This relation is a direct result of the Legendre transformation of the $M(T)$ relation of the LK formula[49], and the detailed derivation is in the Methods. Note that the function $R_T''(T)$ changes sign near $\frac{2\pi^2 k_B T}{\hbar} = 1.62$, a 180° phase shift of the QOs is expected to be observed in $\lambda_{xx}$ and $\lambda_{xy}$ at elevated temperatures. Experimentally, given the fact that the thermal Hall signal in most materials is already difficult to measure, the exact $R_T''(T)$ temperature dependence of the amplitude of the oscillatory magnetothermal effect has not yet been studied. For example, even though bismuth was one of the earliest materials discovered to exhibit QOs in magnetization, resistivity, and the giant oscillatory

Nernst coefficients, neither the thermal Hall effect nor the QO in the thermal transport properties is observed in bismuth[50,51].

Figure 2a shows the raw data of the QOs in $\lambda_{xy}T$ with a constant value shifting at different $T$ ranging from 3.25 K to 17.15 K. Due to the multiple frequency QOs, a 75 T high-pass filter is applied to only allow through the primary oscillation from the $\delta$ orbit. The oscillation component with a single frequency is plotted in Fig. 2b. Starting from 3.25 K, the oscillation amplitude gradually vanishes at specific $H$ as $T$ increases. The position of these nodes where $R_T''(T)$ changes sign strictly obeys the function $T = \frac{1.62 \hbar e}{2\pi^2 k_B m^*} \mu_0 H$. At elevated $T$, the oscillation appears again, accompanied by a 180° phase flip. Figure 2c shows the 2D plot of the phase sharpness $\psi$ of the observed oscillation. (see Supplementary Note 7 for the definition of $\psi$). The phase diagram is divided into two regions with different colors. At low $T$ and strong $H$, $\psi > 1$, the region is colored orange. At high $T$ and weak $H$, $\psi < 1$, the phase shifts to the opposite, and the area is colored light blue. At the boundary of these two regions, $\psi \approx 1$, which means the oscillation amplitude is ~0 and the phase is difficult to distinguish. The boundary of the phase-shifting is fitted using a straight line in the 2D $H$-$T$ plot (Fig. 2d). As the $T$ increases, the boundary linearly shifts to higher $H$, exactly following the linear relationship $T = \frac{1.62 \hbar e}{2\pi^2 k_B m^*} \mu_0 H$ provided by our model. From the slope of this line, the cyclotron effective mass $m^*$ is estimated to be 0.13 $m_e$, where $m_e$ is the free electron mass. The effective mass revealed by the phase shift is similar to the previously reported result[31,47].

Next, we plot the temperature variance of the oscillation amplitudes $\Delta \lambda_{xx}$ and $\Delta \lambda_{xy}$ in Fig. 3a. Going from the lowest temperature, the oscillation amplitudes show a concave curve. This can be understood by considering that the electron and phonon contribute to $\kappa_{xx}$ and $\kappa_{xy}$ additively. However, the oscillation amplitude of $\lambda_{xx}$ and $\lambda_{xy}$ will be influenced by the non-oscillating phonon thermal conductivity from the matrix inversion. Thus, $\kappa_{xx}$ and $\kappa_{xy}$ should be utilized to investigate the $R_T''(T)$ damping effect to eliminate the contribution from the phonon (see Supplementary Note 2). As shown in Fig. 3b, the purple circles and light blue squares are the calculated ratio between the oscillation amplitude of magnetothermal conductivity in various temperatures and the electrical conductivity in the lowest temperature limit. By doing the fitting for the experimental data using the $R_T''(T)$ function, the temperature smearing of the thermal conductivity QOs which follow the second derivative of $R_T(T)$ is confirmed. Similar to the specific heat QOs in high-temperature superconductors[52,53], the observation of the $R_T''(T)$ temperature dependence and the 180° phase shift also demonstrates an essential signature of the magnetothermal QOs expected for a Fermi liquid.

### The strong violation of the WF law in the thermal Hall QOs of CsV$_3$Sb$_5$

The WF relation can also be checked in the oscillatory components similar to the non-oscillating background. Generally, in the limit of low temperature, strong field, and small effective mass, the phase smearing effect caused by finite temperature can be neglected. Thus, the oscillation of the same Landau level density of states at the Fermi surface has constructive interference and the oscillatory amplitudes $\Delta \kappa_{ij}$ and $\Delta \sigma_{ij}$ should be related by the WF relation when $T \to 0$:

$$\frac{\Delta \kappa_{ij(T \to 0)}}{\Delta \sigma_{ij(T \to 0)}} = L_0 T. \quad (2)$$

At elevated temperatures, $\Delta \kappa_{ij}$ and $\Delta \sigma_{ij}$ are damped by the coefficients $R_T''(T)$ and $R_T(T)$ respectively. However, in Kagome metal CsV$_3$Sb$_5$, the WF law of the QOs is violated. Figure 3b shows the temperature dependence of the ratio between $\Delta \kappa_{ij(T \to 0)}$ and $\Delta \sigma_{ij(T \to 0)}$. From the WF relation, the very low $T$ limit values for the longitudinal and transverse channels should both be $L_0$. In Fig. 3b, the experimental results showed that $\Delta \kappa_{xx(T \to 0)}/\Delta \sigma_{xx(T \to 0)}$ reaches 1.55 $L_0$ and $\Delta \kappa_{xy(T \to 0)}/\Delta \sigma_{xy(T \to 0)} T$ is a even

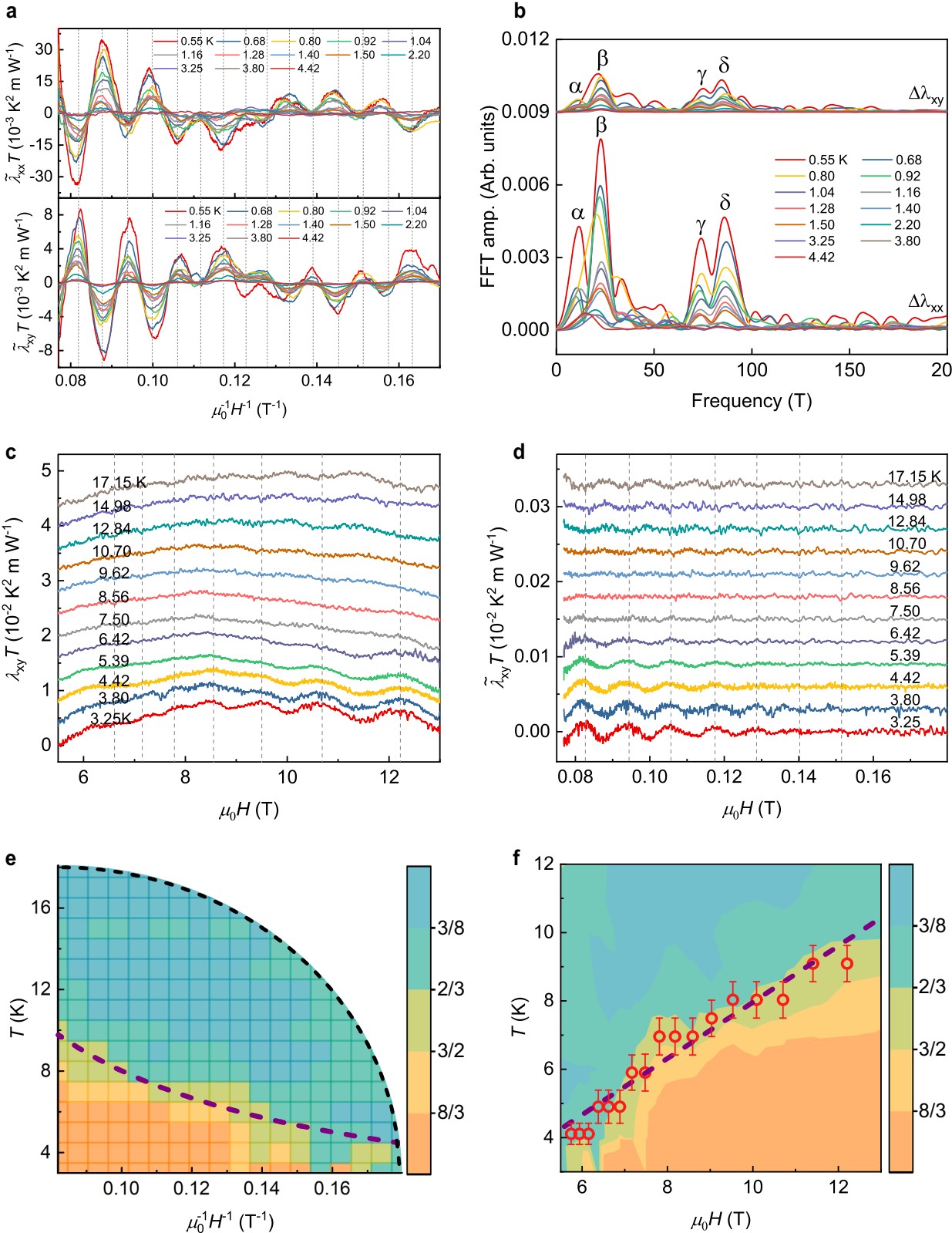

larger value 2.50 $L_0$. By contrast, the background ratio $\kappa_{xy(T\to0)}/\sigma_{xy(T\to0)}T$ has already been shown consistent with the WF law, and the ratio is $L_0$ in low $T$.

Generally, $\kappa_{xx}$ and $\kappa_{xy}$ are related to both electrical conductivity and specific heat. A question is whether the magnetothermal QOs in CsV$_3$Sb$_5$ follow the SdH or dHvA mechanism. It has been shown that with multiple electron and hole bands, the pronounced QOs of the Hall

effect depend on the relative contribution of each band, respectively[54]. However, in multiband Kagome metal CsV$_3$Sb$_5$, the multiband feature only has a secondary influence on the oscillatory components $\kappa_{xx}$ and $\kappa_{xy}$. Figure 3c shows the ratios between $\Delta\kappa_{xy}$ and $\Delta\kappa_{xx}$ of the four smallest pockets in CsV$_3$Sb$_5$. For these different pockets, the ratios are all around the same value and slightly go up in the same trend when $T$ increases, which means the QOs of $\kappa_{xx}$ and $\kappa_{xy}$ follow the dHvA pattern

**Fig. 2 | The quantum oscillations in $\lambda_{xx}$ and $\lambda_{xy}$ and the 180-degree phase flip in $\lambda_{xy}$. a** The oscillatory components $\tilde{\lambda}_{xx}$ and $\tilde{\lambda}_{xy}$ were extracted from $\lambda_{xx}$ and $\lambda_{xy}$ at selected temperatures, obtained after a fifth-order polynomial background subtraction from the raw data. A primary contribution from the delta orbit (-87 T) is marked by the grey dashed line. **b** Fourier transformation of the quantum oscillations at field ranged from 4 T to 13 T, showing four principal frequencies, $F_\alpha = 11$ T, $F_\beta = 25$ T, $F_\gamma = 72$ T, and $F_\delta = 87$ T. **c** Raw data of the quantum oscillation in thermal Hall signal at different $T$ ranging from 3.25 K to 17.15 K. Each curve is shifted with a constant. **d** The oscillatory components were obtained after fifth-order polynomial background subtraction. The 75 T high-pass filter is applied to emphasize the oscillation from the delta orbit (-87 T). As the temperature goes from 3.25 K to 17.15 K at magnetic field $H = \frac{2\pi^2 k_B m^*}{1.62\mu_0 he} T$, the oscillation amplitude passes through

zero, accompanied by a phase reversal. **e** The relative phase $\psi$ of the oscillation in the $T$ vs. $1/H$ phase diagram (see Supplementary Note 7 for the definition of $\psi$). At low $T$ and strong $H$, $\psi > 1$, the region is colored with orange. At high $T$ and weak $H$, $\psi < 1$, the phase becomes opposite, and the area is colored blue. The purple dashed line shows the fitted phase-shifting boundary with function $T = \frac{1.62he}{2\pi^2 k_B m} \mu_0 H$. **f** In the $T$ vs. $H$ phase diagram, the zero amplitude locations of the oscillations are plotted as the red circles. The zeros match the $\psi = 1$ boundary in (**e**), and the error bar is estimated from the broadening of the boundary. The straight purple dashed line is linear fitting of the boundary using the same parameter as Fig. 3c. The slope of this straight line gives the value of $\frac{1.62he}{2\pi^2 k_B m}$, and the cyclotron mass was determined as $0.13m_e$.

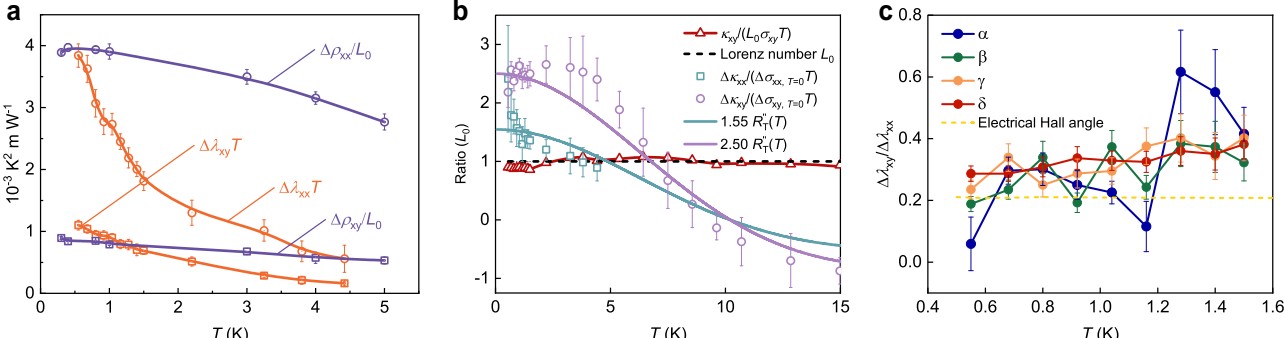

**Fig. 3 | The quantum oscillation amplitude in thermal and thermal Hall resistivity and conductivity. a** The oscillation amplitude $\Delta\lambda_{xx}T$, $\Delta\lambda_{xy}T$, $\Delta\rho_{xx}/L_0$, $\Delta\rho_{xy}/L_0$ as a function of temperature obtained by performing the FFT to the oscillatory components. As $T$ increases, $\Delta\rho_{xx}/L_0$ and $\Delta\rho_{xy}/L_0$ damp much slower than $\Delta\lambda_{xx}T$ and $\Delta\lambda_{xy}T$, indicating the magnetothermal oscillation amplitudes have a different temperature damping factor from the electrical resistivity. **b** To eliminate the contribution from the phonon, $\kappa_{xx}$ and $\kappa_{xy}$ were utilized to investigate the $R_T''(T)$ damping effect. The purple circles and light blue squares show the calculated ratio between the oscillation amplitude of the magnetothermal conductivity in various temperatures and the electrical conductivity in the zero temperature limit. The amplitudes were read directly from Supplementary Fig. 6 at $1/\mu_0 H - 0.082 T^{-1}$. The error bars are estimated from the broadening of the raw data due to the experimental noise. The purple and light blue solid lines are the fittings for the

longitudinal and transverse thermal conductivity oscillation amplitudes using the $R_T''(T)$ function. The fitting results of the oscillation amplitude are $1.55 R_T''(T)$ for the longitudinal thermal conductivity and $2.50 R_T''(T)$ for the transverse thermal Hall conductivity. The experimental results showed that in the zero temperature limit, the oscillation WF ratio $\Delta\kappa_{xx}/\Delta\sigma_{xx}T$ reaches $1.55 L_0$, and $\Delta\kappa_{xy}/\Delta\sigma_{xy}T$ reaches $2.50 L_0$. The background WF ratio $\kappa_{xy}/\sigma_{xy}T$ is plotted using the red triangles. The Lorenz number $L_0$ is shown as the black dashed line. **c** The ratio of the transverse and longitudinal thermal conductivity oscillation amplitudes for the four smallest orbits $\alpha$, $\beta$, $\gamma$, and $\delta$. The ratio is calculated using the FFT amplitude for these different orbits separately. The four different frequencies give nearly the same ratio at different $T$, indicating the oscillations are mainly dominated by the thermodynamic potential rather than the multiband feature. The error bars of (**a**) and (**c**) are estimated by comparing the differences of several common FFT windows.

instead of the SdH pattern. This phenomenon happens when the mean free path is comparable to the sample size or domain size at low $T$[55]. Then, both $\kappa_{xx}$ and $\kappa_{xy}$ are proportional to the electronic specific heat $C$, and their QOs are dominated by the thermodynamic potential (see Supplementary Note 5).

## Discussion

Our findings reveal the QOs in the thermal Hall effect in correlated quantum materials for the first time. The oscillatory thermal Hall signal shows a characteristic 180° phase flip in elevated temperatures, which is the macroscopic effect caused by microscopic quantum phase interference due to the modification of the distribution function for thermodynamic quantities[56]. Meanwhile, the observed temperature variance of the oscillatory components and the phase flip at elevated temperatures is remarkably consistent with the predictions based on the Fermi liquid theory, which is a "smoking gun" for any model. Since the magnetothermal QOs originate from the Landau level density of states (DOS), the $R_T''(T)$ temperature dependence with a characteristic phase flip can be used to identify and investigate the oscillatory behavior of magnetothermal conductivity in a wide range of quantum materials such as single element metals[57–59], quantum spin liquid candidates $\alpha$-RuCl$_3$[1], semimetals[60–62], and high-temperature superconductors[52,53].

On the other hand, The oscillatory WF ratio enables to examine the correlated materials from a different perspective. Usually, the background WF law violation was used to reveal unconventional

quasiparticle dynamics. In terms of the oscillatory WF ratio, the deviation is also highly unconventional. Whereas before claiming the deviation of the oscillatory WF ratio in CsV$_3$Sb$_5$ is related to its electronic structure, it may be that one should consider that this deviation is due to phonon. Recent works suggest that the phonon drag effect may lead to enormously large QOs in the Nernst effect in Weyl semimetals[63] and the significantly enhanced thermal Hall effect in doped SrTiO$_3$[64]. Yet, the QOs in the thermal Hall effect have not been reported in these materials, and the phonon drag effect was not observed in CsV$_3$Sb$_5$ from the thermoelectric measurements. Moreover, in the limit of considerably low $T$ as in this study, the phonon-drag effect is likely to be very small. Thus, we can rule out the possibility that phonons contribute to the thermal Hall QOs.

Since $\kappa$ and $\sigma$ measure the dHvA and SdH QOs, respectively, the observation of the OQs WF law violation reveals different physical manifestations of dHvA and SdH QOs mechanism. Generally speaking, both dHvA and SdH effects are due to the oscillatory density of states (DOS). Nevertheless, the SdH effect has an extra term coming from the scattering between the oscillating states and the total states on the Fermi surface, and the amplitude of the oscillation component of electrical conductivity must be normalized by dividing by the non-oscillatory background[65–67]. The combination of these two factors can lead to the deviation of the SdH QOs amplitudes from the DOS QOs amplitudes, especially when the Landau level orbits are unconventional, such as the existence of magnetic breakdown[66] or strong

correlation effects[67,68]. Indeed, recent studies support the magnetic breakdown effect between conventional orbits and Chern Fermi pockets[48,69] which has the same frequency as the $\delta$ orbit observed in magnetothermal QOs in this paper. Many works also observed the strongly interacting effects with multiple interrelated phases in this system[36,45,46,70–72]. In this sense, the oscillatory magnetothermal effect provides a bridge for the direct comparison between the SdH and dHvA effects and sheds light on how the intricate physics in Kagome lattice leads to novel correlation effects that may contribute to the emergence of unusual density-wave order, electronic nematicity, and putative intertwined orders[35,36,46,70,71].

## Methods

### Experimental setup of $CsV_3Sb_5$ sample in the main text

The $CsV_3Sb_5$ single crystal was synthesized via a self-flux growth method similar to the previous reports[26]. The dimension of the $CsV_3Sb_5$ sample in the main text is 1.8 mm × 1.6 mm × 0.06 mm. The thickness of the sample is quite homogeneous at different positions or along its entire length. The measurement was performed in the Oxford Triton200-10 Cryofree Dilution Refrigerator. The longitudinal and transverse thermal conductivities were measured using a one-heater-three-thermometers technique (Fig. 1a). One end of the sample is thermally anchored (with H74F epoxy of Epoxy Technology)) on the copper heat bath of the probe sample chamber. To measure the longitudinal and transverse temperature differences $\Delta T_x$ and $\Delta T_y$ on the sample, the 5 mil Au wires were first attached to the sample with Ag paint. The positions of the contacts are located near the center of the sample. The sizes of the contacts are smaller than 0.05 mm. Then, a 1 kOhm thin-film resistor heater and three Lakeshore Cryotronics RX102A thermometers (1 kOhm) were thermally attached to the Au wires with Ag paint. Low thermal conductivity Lakeshore Manganin wires of 1-mil diameter were connected from the contact on the heater and thermometers to the copper wires extended to the exterior. The steady heating power $P$ applied to the warm edge of the sample is always set to make sure the temperature change on the sample is less than 10% of the environment temperature. To ensure the magnetic field is perfectly perpendicular to the sample, the sample was first put on top of two Cu blocks with the same thickness. The copper blocks were then removed after the sample was firmly anchored.

The longitudinal and transverse temperature differences $\Delta T_x$ and $\Delta T_y$ were read by Lakeshore Cryotronics RX102A thermometers. The $H$-field was varied continuously and slowly (0.05 T/min) during the measurement at a stable temperature. The results were also checked to be consistent with the stepped-field method to avoid the eddy current or other heating effects. The field dependence of the measured signal was plotted as thermal and thermal Hall resistivities to prevent the error in doing the matrix inversion. The thermal and thermal Hall resistivities were measured at fixed $T$ with $H\|z$ and $-\nabla T\|x$. The temperature stability of the heat bath is controlled to be smaller than $10^{-4}$ K using the monitoring thermometer. There is no $T$ gradient on the heat bath since the Cu heat bath is larger than the sample. Thus, the possible error induced by the thermal Hall effect of Cu can be ignored.

We note that in $CsV_3Sb_5$ various experimental probes revealed potential micro-size domain at low $T$ due to the CDW supercells[73–75]. A pioneering Focus-Ion-Beam work[72] milled hexagon-shaped micro-structures with a size of 10 $\mu$m to search for electrical transport anisotropy. However, given the mm sizes of samples used in the thermal transport measurement, the thermal transport properties are the averages of many domains.

### The differential method and thermometer calibration

The experimental data were taken in the DC static heater method. The resistance of the thermometers is quite stable and will not drift when sweeping the magnetic field. The results were checked to be consistent with the square-wave and sine-wave heating power measurements. The resistances of the thermometers $R_{T1}$, $R_{T2}$, and $R_{T3}$ were measured by the Lock-in amplifiers (SR865) at frequency $f$ - 13 Hz. ($T_1$, $T_2$, and $T_3$ are measured at the contacts shown in Fig. 1a) To get a better signal-of-noise ratio and further reduce the temperature fluctuation of the environment, the transverse signal was measured in a differential method: the resistance difference $\Delta R$ between the $R_{T2}$ and $R_{T3}$ was detected by a Lock-in amplifier after being amplified by a differential amplifier (×100). Then, $R_{T3}$ can be obtained by calculating $R_{T2} + \Delta R/100$. During the measurement, the amplifier was placed into a box with high permeability, and the wires were wrapped together by Al foils to reduce the $H$-field induced noise pick-up. To convert the exact temperature from the measured resistance, all the thermometers are calibrated as a function of temperature and magnetic field[76]. With the above procedures, the resolution of the transverse temperature difference was successfully reduced to the range of $10^{-5}$–$10^{-4}$ K while at the heat bath temperature $T = 1$ K (see Supplementary Fig. 4d).

## Data availability

The data generated in this study have been deposited in the OSF.io database under accession code https://osf.io/fbtzu/.

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

## Acknowledgements

The work at the University of Michigan is primarily supported by the National Science Foundation under Award No. DMR-2004288 and No.DMR- 2317618 (electrical transport and thermal transport properties measurements) to Kuan-Wen Chen, Dechen Zhang, Guoxin Zheng, Aaron Chan, Yuan Zhu, Kaila Jenkins, and Lu Li. The supporting magnetization measurements at the University of Michigan acknowledge the support by the Department of Energy under Award No. DE-SC0020184. The University of Science and Technology of China (USTC) provided the crystals in 2021 and U-M generated the data for this project using the provided crystals in 2022. The crystal growth work at the University of Science and Technology of China is supported by the National Natural Science Foundation of China under Grant No. 11888101 and by the Strategic Priority Research Program of the Chinese Academy of Sciences under Grant No. XDB25000000.

## Author contributions

D.Z., K.-W.C., G.Z., Y.Z., A.C., K.J., Z.X., and L.L. performed the thermal transport measurements, electrical transport, and magnetization measurements. D.Z., K.-W. C., Z.X., and G.Z. developed the setup for the measurements. F.Y., M.S., J.Y., and X.C. grew the high-quality single crystalline samples. D.Z. and L.L. analyzed the data and prepared the manuscript.

## Competing interests

The authors declare no competing interests.
