## [Peer Review File · Nature Communications]

REVIEWER COMMENTS

Reviewer #1 (Remarks to the Author):

The authors report electrical- and thermal-transport measurements done in CsV₃Sb₅. They have successfully measured quantum oscillations (QOs) of the thermal Hall conductivity (k_{xy}), as well as those in the electrical Hall conductivity (σ_{xy}), in the correlated kagomé metal. They further find that the QOs in k_{xy} exceeds the value given by the WF law to the QOs in σ_{xy} . They also argue that these results are useful in terms of studying possible QOs in insulating materials.

The experimental results reported in this paper have been very carefully and systematically collected. It is always challenging to detect QOs in correlated materials that usually require very sensitive measurements done in very low temperatures and high magnetic fields on a very good sample. It is also challenging to detect k_{xy} that is much smaller than k_{xx} . Therefore, it is a remarkable achievement to detect QOs in k_{xy} . As shown by the long “Methods” section, the authors have also carefully analyzed their results from a variety of perspectives including possible phonon contribution to QOs in k_{xy} . It is indeed intriguing finding that the amplitude of the QOs in k_{xy} exceeds that given by the WF law, whereas the background non-oscillating part of k_{xy} fulfills the WF law. The results reported in this manuscript would have a significant impact in the broad research area of correlated electrons, as well as an implication on recent reports about possible QOs measured in insulating materials such as the Kitaev candidate α -RuCl₃ and Kondo insulators SmB₆ and YbB₁₂.

My only concern is the insufficient description about the sample characterization. Information usually relevant to QOs, such as ρ_0 , RRR, or the sharpness of T_c , should be indicated for both sample #1 and #2, so that the readers, seeking to perform similar QO measurements in a different material, can gain insight into the quality of the samples needed to measure quantum oscillations in the thermal Hall measurements.

I would recommend the publication of this manuscript if this concern is well addressed.

Another, rather editorial, comment is that it is quite frustrating to find which part in Methods is referred by “see Methods” in the main text, because Methods section is very long. Please clarify this.

Reviewer #2 (Remarks to the Author):

In this manuscript, D. Zhang et al. report experimental findings of oscillating thermal Hall conductivity for Kagome metal CsV₃Sb₅. From the experimental point of view, it is a beautiful study with wonderful results. As mentioned in the paper, since the quantum oscillation of the thermal Hall conductivity is typically invisible due to the tiny effect, the demonstration of the quantum oscillation of the thermal Hall conductivity in Kagome metal CsV₃Sb₅ provides the key breakthrough to understanding the physical phenomena in the intriguing Kagome metal.

However, I think the way the paper is written is not always appropriate, including a few wrong statements and too many confusing expressions. Moreover, I have the feeling these results could be further exploited to better support the conclusions. So, I would not recommend the publication of the paper in its present form. The results are very interesting, and I think that this paper could be reconsidered if the authors rewrite the manuscript with a better form. For example, the authors need to explain the experimental results first and then provide a discussion with the possible interpretations one by one. And I list the points below.

In the sentence “the capacity of charged quasiparticles to transport heat and charge is governed by the universal WF law.” on page 3, what is the meaning of “the capacity”?

In the sentence “Indeed in CsV₃Sb₅, we found the low-temperature oscillation amplitude of thermal hall conductivity is enhanced by a factor of 2.5 compared with that in electrical Hall conductivity, which cannot be probed by the conventional WF ratio.” on page 3, it seems to compare the experimentally measured amplitudes of the thermal Hall conductivity and the expected amplitude of the thermal Hall conductivity by using the electrical Hall conductivity and WF ratio. Clarify the paragraph without confusion.

In the sentence of “The transverse temperature difference ΔT_{xy} is generated by the longitudinal temperature difference along the x-direction and the magnetic field along the z-direction.” in Figure 1(a) caption, check the orientation. As far as I know, ΔT_{xy} should be generated by the transverse difference along the y-direction.

In Figure 1(c) caption, they describe, “The pink and blue curves with open squares and circles are the electronic thermal conductivity at 0 T and 13 T calculated according to the WF law respectively.”. But, in Figure 1(c), the $L_0 \sigma_{xx} T$ at 13 T is squares, not circles.

The unit of κ_{xy}/T of Figure 1(f) should be [$W K^{-2} m^{-1}$] not [$W K^{-1} m^{-2}$].

In the last sentence of the Figure 1 caption, it should be better that the authors provide the employed Lorenz number L_0 .

Provide temperature dependence of magnetization of the sample across T_c .

In the second paragraph of page 4, “As shown in Fig. 1d and Fig. 1e, ...” should be changed to “As shown in Fig. 1e, ...”

In the last sentence of the second paragraph on page 4, "... above 30 K .." should be changed to "... up to 30 K ...".

In the fourth paragraph on page 4, "... those in the SdH oscillation⁴⁹." is the first mention of the Shubnikov-de Haas oscillation. So, it should be described as "... those in the Shubnikov-de Haas (SdH) oscillation⁴⁹." with the full description, not the abbreviation.

FFT results of the extended Data Fig. 4 (b) should be presented as the main figure with the main text, not as the extended data figure.

In the first paragraph of page 5, "... the well-established LK ..." is the first mention of the Lifshitz-Kosevich formula. So it should be described as "... the well-established Lifshitz-Kosevich (LK) ...", not the abbreviation.

In Figure 2(b), the caption, "... at a specific magnetic field, ..." provides a value/values of the specific magnetic field.

In Figure 2(c) caption, which oscillation does the relative phase refer to as the in-phase oscillation? What is the unit of the phase?

Provide the physical parameters to correspond to the color information in Figures 2(b) and 2(c).

What is the difference between the phase sharpness parameter and the relative phase? What is the physical meaning of the phase sharpness parameter?

An intensive study on the quantum oscillations in the thermal conductivity in Kagome metal CeV₃Sb₅ is reported. The authors performed a careful measurement and analysis on the longitudinal thermal conductivity and thermal Hall conductivity. They found a 180-degree phase flip of quantum oscillations, as a special feature of thermal transport properties. Furthermore, the violation of Wiedemann Frantz (WF) law in the amplitude of the quantum oscillations in the Hall channel is demonstrated after establishing that the WF-law is valid for the non-oscillating components. I found that the observation of the quantum oscillation in the thermal Hall effect thanks to their high-precision measurements is remarkable, and that the analysis is carefully and properly conducted. I'm not so convinced with their discussion on the reason of the enhancement of the WF-ratio, however, the technique of the oscillatory thermal Hall effect would attract the attention of the broad readers of Nature Communications. I recommend to publication of this manuscript to Nature Communications after the proper revisions.

1. Why is the Wiedemann-Franz law violated only in the amplitude of the quantum oscillation, but not in the value of smooth background of κ_{xy} obeys? The LK formula itself seems to be robust, given the fact that the temperature dependence is well explained by the second derivative of $R(T)$, but why does only the amplitude change?
2. Is it possible to derive the WF-law in quantum oscillations from LK formula? It will make their claim of the violation of WF-law clearer.
3. What is special about CsV₃Sb₅ that enables the observation of the quantum oscillations in κ_{xy} ? The authors mentioned that “the thermal Hall signal in most materials is already difficult to measure” and “neither the thermal Hall effect nor the QO in the thermal transport properties of bismuth is too small to be detected” in the main text, however, the reason why CsV₃Sb₅ is suitable for their measurement is not so clearly written. The comparison of the observed size or the upper boundary of the thermal Hall signal in various materials may be useful for the discussion.
4. Is it possible to demonstrate the 180-degree phase flip of quantum oscillations using the longitudinal thermal conductivity κ_{xx} ? It will support the claim of Fig. 2.
5. Error bar should be added to Fig. 3C.
6. In Extended Data Fig. 11, the left label should be $\lambda_{xy} T$, not $\lambda_{xx} T$.

One-by-one response

Thank you very much for sharing the review reports and allowing us to address the questions. We are also very grateful for the constructive feedback from the referees. We have carefully read the comments from the reviewers and made our total effort to address them by performing additional measurements, revising the main text, figures, and supplementary materials, and providing in-depth explanations of concepts/terminologies. In this submission, the reviewers' comments are quoted and in black, and we provide the point-by-point response to all reviewers' comments in blue. The corresponding revised contents are highlighted in red in the revised main text and Supplementary Materials.

“Reviewer #1 (Remarks to the Author):

“The authors report electrical- and thermal-transport measurements done in CsV₃Sb₅. They have successfully measured quantum oscillations (QOs) of the thermal Hall conductivity (k_{xy}), as well as those in the electrical Hall conductivity (σ_{xy}), in the correlated kagomé metal. They further find that the QOs in k_{xy} exceeds the value given by the WF law to the QOs in σ_{xy} . They also argue that these results are useful in terms of studying possible QOs in insulating materials.

“The experimental results reported in this paper have been very carefully and systematically collected. It is always challenging to detect QOs in correlated materials that usually require very sensitive measurements done in very low temperatures and high magnetic fields on a very good sample. It is also challenging to detect k_{xy} that is much smaller than k_{xx} . Therefore, it is a remarkable achievement to detect QOs in k_{xy} . As shown by the long “Methods” section, the authors have also carefully analyzed their results from a variety of perspectives including possible phonon contribution to QOs in k_{xy} . It is indeed intriguing finding that the amplitude of the QOs in k_{xy} exceeds that given by the WF law, whereas the background non-oscillating part of k_{xy} fulfills the WF law. The results reported in this manuscript would have a significant impact in the broad research area of correlated electrons, as well as an implication on recent reports about possible QOs measured in insulating materials such as the Kitaev candidate α -RuCl₃ and Kondo insulators SmB₆ and YbB₁₂.

“My only concern is the insufficient description about the sample characterization. Information usually relevant to QOs, such as ρ_0 , RRR, or the sharpness of T_c , should be indicated for both sample #1 and #2, so that the readers, seeking to perform similar QO measurements in a different material, can gain insight into the quality of the samples needed to measure quantum oscillations in the thermal Hall measurements.

“I would recommend the publication of this manuscript if this concern is well addressed.”

Response: We thank Reviewer #1 for his/her valuation of the manuscript. We also appreciate this suggestion. The sample ρ_0 is 1.495 $\mu\Omega$ cm for sample #1 and 2.04 $\mu\Omega$ cm for sample #2. The residual resistivity ratio RRR value is calculated using the electrical conductivity ratio at 5 K and 300 K, the value of $\frac{\sigma_{5K}}{\sigma_{300K}}$ is 59.7 for sample #1 and 49.4 for sample #2. Following, please find the figures of the temperature dependence of the sample resistivities, which are also included in the revised Fig. S1 in the supplement.

Fig. R1. Temperature dependence of resistivity of Sample #1 and #2 in the range from 1.5 K to 300 K (Panel a) and from 1.5 K to 5 K (Panel b). This figure is Panel a and Panel b of the new Fig. S1 in the supplement.

“Another, rather editorial, comment is that it is quite frustrating to find which part in Methods is referred by “see Methods” in the main text, because Methods section is very long. Please clarify this.”

Response: Yes, this is a good point. We have moved the long Methods section to the Supplementary Text and Figures. We have inserted the exact part number for this reference to the Supplementary Materials.

“Reviewer #2 (Remarks to the Author):

“In this manuscript, D. Zhang et al. report experimental findings of oscillating thermal Hall conductivity for Kagome metal CsV₃Sb₅. From the experimental point of view, it is a beautiful study with wonderful results. As mentioned in the paper, since the quantum oscillation of the thermal Hall conductivity is typically invisible due to the tiny effect, the demonstration of the quantum oscillation of the thermal Hall conductivity in Kagome metal CsV₃Sb₅ provides the key breakthrough to understanding the physical phenomena in the intriguing Kagome metal. However, I think the way the paper is written is not always appropriate, including a few wrong statements and too many confusing expressions. Moreover, I have the feeling these results could be further exploited to better support the conclusions. So, I would not recommend the publication of the paper in its present form. The results are very interesting, and I think that this paper could be reconsidered if the authors rewrite the manuscript with a better form. For example, the authors need to explain the experimental results first and then provide a discussion with the possible interpretations one by one. And I list the points below.

“In the sentence “the capacity of charged quasiparticles to transport heat and charge is governed by the universal WF law.” on page 3, what is the meaning of “the capacity”?”

Response: We thank the reviewer for catching this point. The word “capacity” should be “ability”. This point is also addressed in the revised manuscript.

“In the sentence “Indeed in CsV₃Sb₅, we found the low-temperature oscillation amplitude of thermal hall conductivity is enhanced by a factor of 2.5 compared with that in electrical Hall conductivity, which cannot be probed by the conventional WF ratio.” on page 3, it seems to compare the experimentally measured amplitudes of the thermal Hall conductivity and the expected amplitude of the thermal Hall conductivity by using the electrical Hall conductivity and WF ratio. Clarify the paragraph without confusion.”

Response: Again, we thank the reviewer for pointing out this issue.

The Wiedemann-Franz (WF) law states that in a Fermi liquid, the quasiparticles that transport charge also carry heat at the same time. Therefore, in most conventional metals the ratio between the electronic thermal conductivity (κ_e) and the product of electrical conductivity (σ) and temperature (T) is a proportionality constant called the WF ratio $L = \kappa_e/\sigma T$, typically not very different from the Lorenz number $L_0 = \left(\frac{\pi^2}{3}\right)\left(\frac{k_B}{e}\right)^2 = 2.44 \times 10^{-8} \text{W ohm K}^{-2}$, (where k_B is the Boltzmann constant and e is the electron charge). To test the WF law, the magnitude of the thermal Hall conductivity background or the oscillation amplitude should be compared with the electrical Hall conductivity background, or the oscillation amplitude multiplied by the Lorenz number and the absolute temperature.

Based on the reviewer’s suggestion, we have revised the sentence as “Indeed in CsV₃Sb₅, we found the low-temperature oscillation amplitude of thermal hall conductivity is enhanced by a factor of 2.5 compared with that in electrical Hall conductivity multiplied by the Sommerfeld value L_0 and the absolute temperature T , which cannot be explained by the conventional WF ratio.

“In the sentence of “The transverse temperature difference ΔT_{xy} is generated by the longitudinal temperature difference along the x-direction and the magnetic field along the z-direction.” in Figure 1(a) caption, check the orientation. As far as I know, ΔT_{xy} should be generated by the transverse difference along the y-direction.”

Response: Thank you for pointing this out. The word “generated” here means with the longitudinal temperature difference applied along the x-direction and the magnetic field applied along the z-direction, we can detect the transverse temperature difference ΔT_{xy} . The transverse temperature difference ΔT_{xy} just means the transverse temperature difference along the y-direction. To clear up the misunderstanding, we changed the label of the transverse temperature difference ΔT_{xy} to ΔT_y .

“In Figure 1(c) caption, they describe, “The pink and blue curves with open squares and circles are the electronic thermal conductivity at 0 T and 13 T calculated according to the WF law respectively.”. But, in Figure 1(c), the $L_0 \sigma_{xx} T$ at 13 T is squares, not circles.”

Response: Thanks for pointing it out. The sentence is revised to “The pink and blue curves with open squares are the electronic thermal conductivity at 0 T and 13 T calculated according to the WF law, respectively.”

“The unit of κ_{xy}/T of Figure 1(f) should be $[W K^{-2} m^{-1}]$ not $[W K^{-1} m^{-2}]$.”

Response: Good catch. We changed it in the revision.

“In the last sentence of the Figure 1 caption, it should be better that the authors provide the employed Lorenz number L_0 .”

Response: Thanks for pointing it out. We revised our manuscript accordingly.

“Provide temperature dependence of magnetization of the sample across T_c .”

Response: Following is the magnetic susceptibility curve of the sample, which is included in the revised Fig. S1 in the supplement.

Fig. R2. The low-field magnetization curve of CsV_3Sb_5 showing the Meissner effect. This figure is Panel c of the new Fig. S1 in the supplement.

In the second paragraph of page 4, “As shown in Fig. 1d and Fig. 1e, ...” should be changed to “As shown in Fig. 1e, ...”

Response: Thanks for pointing it out. We revised our manuscript accordingly.

“In the last sentence of the second paragraph on page 4, “... above 30 K ..” should be changed to “... up to 30 K ...”.

Response: Thanks for pointing it out. We revised our manuscript accordingly.

“In the fourth paragraph in page 4, “... those in the SdH oscillation⁴⁹.” is the first mention of the Shubnikov-de Haas oscillation. So, it should be described as “... those in the Shubnikov-de Haas (SdH) oscillation⁴⁹.” with the full description, not the abbreviation.”

Response: Thanks for pointing it out. We revised our manuscript accordingly.

“FFT results of the extended Data Fig. 4 (b) should be presented as the main figure with the main text, not as the extended data figure.”

Response: Thanks for pointing it out. We present the extended Data Fig. 4 (b) as the main figure in the main text.

“In the first paragraph of page 5, “... the well-established LK ...” is the first mention of the Lifshitz-Kosevich formula. So it should be described as “... the well-established Lifshitz-Kosevich (LK) ...”, not the abbreviation.”

Response: Thanks for pointing it out. We revised our manuscript accordingly.

“In Figure 2(b), the caption, “... at a specific magnetic field, ...” provides a value/values of the specific magnetic field.”

Response: This is a good point and would need a better clarification. According to the temperature dependence of the oscillation amplitude of the magnetothermal effect derived in the manuscript, the well-established LK relation for magnetization and resistivity is replaced by the second derivative of $R_T(T)$ in magnetothermal effect. The function $R_T''(T)$ changes sign near $\frac{2\pi^2 k_B T}{\hbar} = 1.62$, which accounts for the 180-degree phase shift of the oscillation as T increases.

Therefore, we considered the reviewer’s suggestion and revised the caption in Figure 2(b) to “As the temperature T goes from 3.25 K to 17.15 K at magnetic field $H = \frac{2\pi^2 k_B m^*}{1.62 \mu_0 \hbar e} T$, the oscillation amplitude passes through zero, accompanied by a phase reversal.”

“In Figure 2(c) caption, which oscillation does the relative phase refer to as the in-phase oscillation? What is the unit of the phase?”

Response: We thank Reviewer #2 for this question on which oscillation the relative phase refers to as the in-phase oscillation and the unit of the phase. We make these definitions in Supplementary Text Section VIII. To clarify, we refer to the Supplementary Text Section VIII: Definition and calculation of the phase sharpness parameter ψ in Figure 2(c) caption.

In the Supplementary Text Section VIII, we add some explanations for the in-phase oscillation. “In-phase oscillation” means it is in phase with the oscillatory components $\Delta\lambda_{xy,\delta}T$ of the δ orbit at lowest T (0.55 K, after the 75 T HPF is applied), and strong magnetic field H . At higher T and weaker H , when the phase of the oscillation becomes opposite, it is called “out-of-phase.”

Since the magnetothermal quantum oscillation phase can either be in-phase at the lowest T and strong H or out-of-phase at higher T and weaker H , it is a binary variable. We define and calculate the phase sharpness parameter ψ to tell whether the oscillation is in-phase or out-of-phase.

“Provide the physical parameters to correspond to the color information in Figures 2(b) and 2(c). What is the difference between the phase sharpness parameter and the relative phase? What is the physical meaning of the phase sharpness parameter?”

Response: We thank Reviewer #2 for this question. We make these definitions in Supplementary Text Section VIII. To clarify, we refer to the Supplementary Text Section VIII: Definition and calculation of the phase sharpness parameter ψ in Figure 2(b) and 2(c) caption. The different colors show different values for the phase sharpness parameter ψ . We define and calculate the phase sharpness parameter ψ to tell whether the oscillation is in-phase or out-of-phase. In contrast, the relative phase determines whether the oscillation is in-phase or out-of-phase, with the oscillation of the δ orbit at lowest T and strong magnetic field H .

“Reviewer #3 (Remarks to the Author):

“An intensive study on the quantum oscillations in the thermal conductivity in Kagome metal CeV_3Sb_5 is reported. The authors performed a careful measurement and analysis on the longitudinal thermal conductivity and thermal Hall conductivity. They found a 180-degree phase flip of quantum oscillations, as a special feature of thermal transport properties. Furthermore, the violation of Wiedemann Frantz (WF) law in the amplitude of the quantum oscillations in the Hall channel is demonstrated after establishing that the WF-law is valid for the non-oscillating components. I found that the observation of the quantum oscillation in the thermal Hall effect thanks to their high-precision measurements is remarkable, and that the analysis is carefully and properly conducted. I’m not so convinced with their discussion on the reason of the enhancement of the WF-ratio, however, the technique of the oscillatory thermal Hall effect would attract the attention of the broad readers of Nature Communications. I recommend to publication of this manuscript to Nature Communications after the proper revisions.

“1. Why is the Wiedemann-Franz law violated only in the amplitude of the quantum oscillation, but not in the value of smooth background of κ_{xy} obeys? The LK formula itself seems to be robust, given the fact that the temperature dependence is well explained by the second derivative of $R(T)$, but why does only the amplitude change? “

Response: This is a great point. While we do not have an exact answer. We suspect the violation only in the oscillation amplitude reveals the unconventional nature of the electronic state of CsV_3Sb_5 . The validity of the Wiedemann-Franz law for the background of κ_{xy} only requires the same thermal and electrical transport relaxation time. Our analysis in the main text shows the QOs of κ_{xy} follow the dHvA pattern instead of the SdH pattern, which happens when the mean free path is comparable to the sample size or domain size at low T . After excluding the phonon contribution to the thermal Hall QOs, we attribute the observation of the QOs WF law violation to different physical manifestations of dHvA and SdH QOs mechanism, especially when the Landau level orbits are also influenced by the existence of magnetic breakdown or strong correlation. The detailed analysis is added in the last two paragraphs of the discussion section in the main text.

“2. Is it possible to derive the WF-law in quantum oscillations from LK formula? It will make their claim of the violation of WF-law clearer. “

Response: We thank Reviewer #2 for this question.

First, the oscillation in resistivity is more related to the oscillations of the density of states. The validity of the WF law for quantum oscillations requires that both the dHvA oscillations (in magnetization and thermal transport) and the SdH oscillations (in resistivity) are due to the same Landau level density of states at the same location on the Fermi surface. Then, all carrier parameters observable via both effects will be identical, and the WF law of the oscillation amplitudes of thermal and electrical transport will follow the WF law of the background.

3. What is special about CsV₃Sb₅ that enables the observation of the quantum oscillations in κ_{xy} ? The authors mentioned that “the thermal Hall signal in most materials is already difficult to measure” and “neither the thermal Hall effect nor the QO in the thermal transport properties of bismuth is too small to be detected” in the main text, however, the reason why CsV₃Sb₅ is suitable for their measurement is not so clearly written. The comparison of the observed size or the upper boundary of the thermal Hall signal in various materials may be useful for the discussion.

Response: This is a great point too. Please let us start by summarizing the key points. The relatively large (not fully canceled) MR certainly helps. The observation benefits from the relatively clean sample, which means large Hall and thermal Hall angles, while the dominating features of quantum oscillations come from small orbits. Developing the bridge technique in electronics also helps make the observation possible.

Then, we explain the reasons in detail. In correlated materials, the oscillation amplitude is often damped by phase smearing due to sample inhomogeneity and the finite relaxation time due to electron scattering. As a result, the measurement of quantum oscillations requires high quality single crystal, low temperature, and strong enough magnetic fields.

However, the measurement of quantum oscillations in the thermal transport properties of CsV₃Sb₅ is more challenging. For example, when the cold finger temperature is at 1 K, the largest temperature drop applied on the sample is less than 0.1 K. Then, the thermal Hall signal is ~ 0.01 K, given that the thermal Hall angle is in the order of 0.1 at 14 T field. The oscillation amplitude of κ_{xy} is $\sim 5\%$ of the overall κ_{xy} signal. Thus the quantum oscillation amplitude in κ_{xy} is in the order of 0.5 mK, as shown in Extended Data Fig. 3 (d). Usually, κ_{xy} is measured in a dilution refrigerator using RuO₂ chip thermometers, the resolution of which is only 0.5 mK at 1 K. Thus, using traditional chip thermometers to measure the absolute temperatures on the sample directly is unlikely to reveal any quantum oscillations in κ_{xy} . So far, even similar oscillations of thermal conductivity have been poorly investigated. To improve the resolution of the transverse thermal Hall signal, we measured the transverse signal using a differential method: instead of measuring the resistance of each thermometer independently, we measured the resistance difference of the two transverse thermometers using the self-designed differential amplifier. The output signal is the resistance difference of the transverse channel with 100 times amplification. This technique breakthrough improves the resolution of the thermal Hall signal from 0.5 mK to 0.01 mK when the sample temperature is 1 K. Moreover, this differential method greatly reduces the noise from the cold finger temperature fluctuation (usually controlled in the order of 0.1 mK) or the external field inhomogeneity, which affects the two transverse thermometers simultaneously.

We also note that many simple metals, such as noble metals and alkali metals, are not suitable for observing the quantum oscillations in κ_{xy} , even when they have a large Hall angle in the order of 1-10 at $B = 1$ T. The noble metals and alkali metals have large Fermi surfaces with high quantum oscillation frequencies, which makes it impracticable to observe the oscillations. Compared with these metals, the orbit size in CsV_3Sb_5 has suitable Landau level spacing to observe the quantum oscillations and the phase inversion. Also, in pure metals, the oscillations are usually small and challenging to detect due to the complicated many-sheeted Fermi surfaces, the large quantum number n , and the large conductivity of the high-purity sample.

“4. Is it possible to demonstrate the 180-degree phase flip of quantum oscillations using the longitudinal thermal conductivity κ_{xx} ? It will support the claim of Fig. 2. “

Response: We thank Reviewer #3 for this question, and this is a great point too. We measured the longitudinal thermal conductivity again, and the result is plotted in Fig. R3(a) and Fig. S 12 in the revised supplement. To analyze the 180-degree phase flip of quantum oscillation using κ_{xx} , we first extracted the oscillatory components of the δ orbit (~ 87 T) using fifth-order polynomial background subtraction, and then we applied the 75 T high pass filter. According to Fig. R3b, the phase flip of quantum oscillations in κ_{xx} happens near $\frac{2\pi^2 k_B T}{\hbar} = 1.62$, which supports the phase flip in κ_{xy} as shown in Fig. 2d and Fig. S 11.

Fig. R3. The raw data of the thermal conductivity (Panel a) and the background-subtracted one (Panel b) show the phase flip. This figure and the detailed explanation are included in Fig. S 12 in the revised supplement.

“5. Error bar should be added to Fig. 3C. “

Response: We thank Reviewer #3 for this question. The ratio $\Delta\kappa_{xy}/\Delta\kappa_{xx}$ for the four smallest orbits $\alpha, \beta, \gamma, \delta$ is calculated using the FFT amplitude for these different orbits separately. The value of the FFT amplitude is obtained by reading the peak height of specific frequency oscillation using the Hanning FFT window. We first perform the FFT transformation using different FFT windows (Hanning, Hamming, Blackman, Flat top, Bohman, Parzen, Welch, Kaiser, etc.), and

then estimate the error bar of $\Delta\kappa_{xy}$ and $\Delta\kappa_{xx}$ using the difference of these methods. The error of the ratio is calculated using:

$$\left(\frac{\sigma_{\Delta\kappa_{xy}/\Delta\kappa_{xx}}}{\overline{\Delta\kappa_{xy}/\Delta\kappa_{xx}}}\right)^2 = \left(\frac{\sigma_{\Delta\kappa_{xy}/\Delta\kappa_{xx}}}{\frac{\Delta\kappa_{xy}}{\Delta\kappa_{xx}}}\right)^2 = \left(\frac{\sigma_{\Delta\kappa_{xy}}}{\overline{\Delta\kappa_{xy}}}\right)^2 + \left(\frac{\sigma_{\kappa_{xx}}}{\overline{\kappa_{xx}}}\right)^2,$$

where $\sigma_{\Delta\kappa_{xy}/\Delta\kappa_{xx}}$ means the error of the ratio, $\sigma_{\Delta\kappa_{xy}}$ means the error of $\Delta\kappa_{xy}$ estimated using different FFT window, $\sigma_{\Delta\kappa_{xx}}$ means the error of $\Delta\kappa_{xx}$, and $\overline{\Delta\kappa_{xy}/\Delta\kappa_{xx}}$ means the value obtained using the Hanning FFT window.

Fig. R4. The raw thermal Hall angle and the error bars are included. The figure is updated in the revision as Fig. 3c.

“6. In Extended Data Fig. 11, the left label should be $\lambda_{xy} T$, not $\lambda_{xx} T$. “

Response: Thanks for pointing it out. We revised our manuscript accordingly.

Additional experiments:

We measured the temperature dependence of resistivity of Sample #1 and #2 in the range from 1.5 K to 300 K. The results are also included in the Supplement as Fig. S1.

We measured the quantum oscillations in the longitudinal thermal conductivity, and the result is plotted in Fig. S 12.

temperature dependence of magnetization of the sample across T_c , and the result is included in the Supplement as Fig. S1.

Summary of manuscript revisions:

Changes in the Main text

(1). Original (Page 3):

The capacity of charged quasiparticles

Revised into (Page 3):

The ability of charged quasiparticles

(2). Original (Page 3):

Indeed in CsV_3Sb_5 , we found the low-temperature oscillation amplitude of thermal hall conductivity is enhanced by a factor of 2.5 compared with that in electrical Hall conductivity, which cannot be probed by the conventional WF ratio.

Revised into (Page 3):

Indeed in CsV_3Sb_5 , we found the low-temperature oscillation amplitude of thermal hall conductivity is enhanced by a factor of 2.5 compared with that in electrical Hall conductivity multiplied by the Sommerfeld value L_0 and the absolute temperature T , which cannot be explained by the conventional WF ratio.

(3). Original (Page 4):

At elevated temperatures above 30 K

Revised into (Page 4):

At elevated temperatures up to 30 K

(4). Original (Page 4):

The oscillations contain multiple orbits whose frequencies are consistent with those in the SdH oscillation.

Revised into (Page 4):

The oscillations contain multiple orbits whose frequencies are consistent with those in the Shubnikov-de Haas (SdH) oscillation and the de Haas-Van Alphen (dHvA) oscillation.

(5). Original (Page 5):

Well established LK T -dependence for magnetization and resistivity.

Revised into (Page 5):

Well established Lifshitz-Kosevich T -dependence for magnetization and resistivity.

(6). Original (Page 5):

Is replaced by the second derivative of $R_T(T)$ for the temperature dependence.

Revised into (Page 5):

Is replaced by $R_T''(T)$, its second derivative with respect to T .

(7). Original (Page 5):

Which accounts for the 180-degree phase shift of the oscillation ...

Revised into (Page 5):

We rewrote this part: a 180-degree phase shift of the QOs is expected to be observed in ...

(8). Original (Page 6):

We plot the temperature variance of the oscillation amplitudes ...

Revised into (Page 6):

We rewrote this paragraph.

(9). Original (Page 7):The paragraph staring with

Finally, the QO amplitudes in the thermal Hall effect is inconsistent with ...

Revised into (Page 7):

We rewrote this paragraph.

(10). Original (Page 8):

To analyze the amplitude of the oscillatory magnetothermal effect of ...

Revised into (Page 8):

We rewrote this paragraph.

(11). Original (Page 9-11):

The last three paragraphs, which is the discussion section.

Revised into (Page 9-10):

We rewrote the last three paragraphs.

Acknowledgments and Author Contributions

We added the Acknowledgements and Author Contributions in the Main text.

Methods

We moved the paragraph on page 7 “We note that in CsV₃Sb₅ various experimental probes revealed ...” to Methods on Page 18 of the revised manuscript.

Figures and Caption

Fig. 1 (A): We changed the label of the transverse temperature difference $\Delta T_{\{xy\}}$ to $\Delta T_{\{y\}}$.

Fig. 1 (F): In the caption, “The pink and blue curves with open squares and circles” revised to “The pink and blue curves with open squares”

Fig. 1 (F): We changed the unit of κ_{xy}/T to [W K⁻² m⁻¹]. In the caption, we provide the employed Lorenz number L_0 .

Fig. 2 (A, B): added Fig. 2 (A) and (B).

Fig. 2 (D): In the caption, “at a specific magnetic field” revised to “at magnetic field $H = \frac{2\pi^2 k_B m^*}{1.62 \mu_0 \hbar \epsilon} T$ ”

Fig. 3 (C): We added the error bar.

Supplementary Materials (We put most of the Methods part into the Supplementary Materials)

The explanation of the in-phase and out-of-phase oscillation was added in Supplementary Text Section VIII.

Added Fig. S 1.

Added Fig. S 12

REVIEWERS' COMMENTS

Reviewer #1 (Remarks to the Author):

In this revision, the authors have addressed all my concerns raised in the previous report. I also find that both the clarity and quality of the manuscript are greatly improved by addressing all the points given by other referees. I do recommend the publication of this manuscript to Nature Communications.

Reviewer #2 (Remarks to the Author):

Most of my comments have been addressed by the authors. However, I still found some non-systematic typos. For example, in the caption of Figure 1(a), ΔT_{xx} and ΔT_{xy} should be changed to ΔT_x and ΔT_y , respectively.

Reviewer #3 (Remarks to the Author):

The authors carefully addressed all the comments from me and other Referees. The revised manuscript further clarifies the importance of the paper in this research area and stimulates future research; I recommend its publication in Nature Communications.